# The Added Value of Diagnostic and Theranostic PET Imaging for the Treatment of CNS Tumors

**DOI:** 10.3390/ijms21031029

**Published:** 2020-02-04

**Authors:** Ilanah J. Pruis, Guus A. M. S. van Dongen, Sophie E. M. Veldhuijzen van Zanten

**Affiliations:** 1Department of Radiology & Nuclear Medicine, Erasmus MC, 3015 GD Rotterdam, The Netherlands; i.pruis@erasmusmc.nl; 2Department of Radiology & Nuclear Medicine, Amsterdam UMC, Vrije Universiteit Amsterdam, 1081 HV Amsterdam, The Netherlands; gams.vandongen@amsterdamumc.nl; 3Princess Máxima Center for Pediatric Oncology, 3584 CS Utrecht, The Netherlands; 4Department of Pediatrics, Amsterdam UMC, Vrije Universiteit Amsterdam, 1081 HV Amsterdam, The Netherlands

**Keywords:** molecular biology, central nervous system, oncology, CNS tumors, positron emission tomography, PET, molecular imaging, targeted therapy, theranostics, drug development

## Abstract

This review highlights the added value of PET imaging in Central Nervous System (CNS) tumors, which is a tool that has rapidly evolved from a merely diagnostic setting to multimodal molecular diagnostics and the guidance of targeted therapy. PET is the method of choice for studying target expression and target binding behind the assumedly intact blood–brain barrier. Today, a variety of diagnostic PET tracers can be used for the primary staging of CNS tumors and to determine the effect of therapy. Additionally, theranostic PET tracers are increasingly used in the context of pharmaceutical and radiopharmaceutical drug development and application. In this approach, a single targeted drug is used for PET diagnosis, upon the coupling of a PET radionuclide, as well as for targeted (nuclide) therapy. Theranostic PET tracers have the potential to serve as a non-invasive whole body navigator in the selection of the most effective drug candidates and their most optimal dose and administration route, together with the potential to serve as a predictive biomarker in the selection of patients who are most likely to benefit from treatment. PET imaging supports the transition from trial and error medicine to predictive, preventive, and personalized medicine, hopefully leading to improved quality of life for patients and more cost-effective care.

## 1. Background

Since the emerge of molecular biology in the 1930s, the discipline has undergone significant changes, which can be largely attributed to the description of DNA as a double-helical structure in 1953, the accomplishment of the Human Genome Project in 2003, and the rapid development of advanced diagnostic technologies. Over the years, cancer diagnostics evolved from gross and microscopic analysis toward an integrated, morphology, and molecular-based approach, leading to improved understanding of carcinogenesis and disease progression [1]. We now understand that cancer is not a monolithic disease and that a tumor is not a homogeneous mass [2]; fighting cancer not only demands an appreciation of inter-patient variability, but also requires us to outwit the intra-tumoral spatial and temporal heterogeneity. Increasing knowledge of the genetic and molecular make-up of tumor subtypes and subclones also led to the development of numerous potentially effective targeted therapies.

Along with the advent of targeted therapies came companion diagnostics, also known as pharmacodiagnostics or theranostics, which are defined by the U.S. Food and Drug Administration (FDA) as “diagnostic devices or imaging tools that provide information that is essential for the safe and effective use of a corresponding therapeutic product”. Companion diagnostics enable the identification and/or quantification of therapy-related biomarkers, and they are used for the selection of patients likely to benefit from treatment or for the identification of patients likely to be at increased risk for serious side effects [3,4]. Companion diagnostics are a prerequisite for receiving the corresponding therapeutic product, which is exemplified by the human epidermal growth factor receptor 2 (HER2) gene expression assessment by immunohistochemistry (IHC) in patients with breast cancer to determine whether they are eligible for trastuzumab treatment [4]. This is in opposition to complementary diagnostics, for which the FDA recently presented a draft definition being: “tests that identify a biomarker-defined subset of patients that respond particularly well to a drug and aid risk/benefit assessments, but that are not a prerequisite for receiving the drug” [4]. Here, the corresponding therapeutic product has shown benefit for the group of patients as a whole, and the complementary diagnostic test will only inform on enhanced benefits in subgroups, such as for example better response to nivolumab (Opdivo) in patients with advanced non-small cell lung cancer (NSCLC) that show higher protein levels of the immune checkpoint protein programmed death-ligand 1 (PD-L1) [4]. To date, 38 therapeutic products and corresponding diagnostic tests, of which only one imaging device (i.e., FerriScan), has been approved by the FDA based on the significant improvement of objective responses and survival benefits in patients with various non-CNS tumors such as breast cancer (response rate (RR) up to 80.2%), NSCLC (RR up to 65%), and colorectal cancer (RR 57%) [4,5].

As for CNS tumors, based on the improvement of diagnostic technologies, in May 2016, the World Health Organization (WHO) published a revised classification as an update of the 2007 edition [1,6]. For the first time, the WHO uses molecular parameters in addition to histology, which has resulted in the dismissal of a number of entities that are no longer thought to have diagnostic and/or biological value and the appointment of newly recognized neoplasms that should facilitate the development of more effective targeted therapies [6]. However, there are a number of significant limitations in today’s diagnostic and theranostic approaches, of which some relate in particular to CNS tumors. As a crucial one, the amount of (viable) tumor material for ex vivo analyses is usually limited, as the procedures for obtaining biopsy samples or resections are invasive and accompanied by the risk of damaging healthy (neuronal) structures. Not every patient with a newly diagnosed CNS tumor undergoes a biopsy or resection, because Magnetic Resonance Imaging (MRI) is usually conclusive. Moreover, most often only (part of) the primary tumor is sampled, and (distant) metastases are left untouched. In addition, tumor material is generally obtained only once or twice during the disease course, at time of diagnosis and/or disease progression, or in very few cases from post-mortem autopsy. These limitations impede a thorough analysis of both spatial and temporal heterogeneity and treatment effects over time, which may lead to misclassification and reduce the chance of developing therapies that are able to outwit all (adapting) carcinogenic mechanisms. Therefore, and for CNS tumors in particular, non-invasive tools to study cancer biology in vivo are highly appreciated. In recent years, several molecular imaging techniques, notably nuclear techniques such as Positron Emission Tomography (PET), have been developed that conjoin both diagnostic and theranostic applications to directly link molecular biology with molecular diagnosis and molecular targeted therapy [7,8]. In our review, we appreciate these developments at multiple levels, with special consideration for the added value in the challenging field of neuro-oncology, where PET imaging not only can serve as a sensitive diagnostic tool enabling non-invasive studies of tumor characteristics at multiple sites over time, but can also serve as an in vivo theranostic tool guiding drug development and drug delivery studies by display of target expression and target binding behind the assumedly intact blood–brain barrier (BBB) [9,10].

## 2. Advanced Technology and Applicability of Molecular PET Imaging for CNS Tumors

In the last decades, the diagnostic imaging armamentarium has greatly improved, ranging from Computerized Tomography (CT) and MRI to single-photon emission computerized tomography (SPECT) and PET. Since the 1990s, hybrid (i.e., fused) technologies arose such as SPECT-CT, PET-CT, and most recently PET-MR imaging, allowing for combined anatomical and functional studies [11]. Of these technologies, the molecular imaging techniques (i.e., SPECT and PET) provide unique possibilities to non-invasively study the molecular biology of disease by using radioactive ligands, so-called tracers, to bind and identify specific targets. Both SPECT and PET have the sensitivity to study target expression and target binding down to the picomolar level [11]. However, PET imaging offers better spatial resolution and quantification abilities. PET also has higher sensitivity, which is particularly beneficial for patients, as it requires lower radiation doses. A next leap forward herein is the recent introduction of advanced whole-body PET scanners as their larger field of view further increases the sensitivity, resulting in better image quality and more accurate assessment of tracer biodistribution at a low count rate [9,10].

Due to the aforementioned reasons, PET is gaining popularity over SPECT. Today, PET and PET tracers can be used in a diagnostic setting for primary staging of CNS tumors and to determine the effect of therapy. In addition, PET and PET tracers can be used to predict the behavior of radioactive or non-radioactive targeted drugs (i.e., radiopharmaceuticals or pharmaceuticals, respectively). Pharmaceuticals used for theranostic PET purposes are ranging from small chemical molecules to biologicals including small peptides, antibodies, antibody fragments, and antibody conjugates, to nanoparticles and effector cells. In case of radiopharmaceuticals, targeted drugs are first labeled with a positron-emitting radionuclide for use in a diagnostic scouting procedure to assess biodistribution and to allow dosimetric analysis to estimate radiation exposure to tumors and critical normal organs. If these analyses look favorable, the same targeting drug—but now labeled with an alpha (α-) or beta (β^-^) particle-emitting radionuclide—is administered to induce localized DNA double-strand breaks and cell death. Of these, alpha-emitting radionuclides have significantly higher relative biological effectiveness compared to beta-emitting radionuclides [12]. In the literature, PET tracers used for therapeutic purposes are designated “companion diagnostics” or “theranostics” in an exchangeable way, although more recently, the term “theranostic” has become more and more reserved for cases where one single molecular entity is used for both diagnostic as well as for therapeutic purposes, and this is also the way we use the term “theranostic” in the present review. To achieve an optimal prediction of theranostic PET tracers, it is important that the diagnostic and therapeutic compound show similar in vivo behavior (i.e., biodistribution), and therefore appropriate selection of radionuclides and radiolabeling procedures is required [13].

PET tracer development and production is challenging and requires a tailor-made approach [14]. First, a target should be selected that is known to be upregulated or expressed specifically by the tumor [15]. Second, the selected ligand (i.e., the pharmaceutical of interest) has to be (in part) specific for the target and should bind with sufficient affinity to enable detection against the background tracer uptake in normal tissue. Third, the physical half-life of the radionuclide should be compatible with the biological half-life and pharmacokinetic characteristics of the targeted pharmaceutical in the body [16]. Fast kinetic biological or small chemical molecules, such as amino acids and tyrosine kinase inhibitors (TKIs), should be conjugated with corresponding short half-life radionuclides. Monoclonal antibodies and other slow kinetic ligands, on the other hand, dictate the use of PET isotopes with a longer half-life [16]. The most frequently used PET isotopes have a short half-life (i.e., 2 min for oxygen-15 [^15^O], 10 min for nitrogen-13 [^13^N], 20 min for carbon-11 [^11^C], 68 min for gallium-68 [^68^Ga], and 110 min for fluorine-18 [^18^F]), which requires a local hospital-based cyclotron, and quick tracer production and clinical supply; thus, they present logistical challenges. On the other hand, PET isotopes with a long half-life (i.e., 12.7 h for carbon-64 [^64^Cu], 78.4 h for zirconium-89 [^89^Zr], and 4.2 days for iodine-124 [^124^I]) do not meet these logistical challenges but bring along a higher radiation burden that should be taken into account [14].

For each newly developed diagnostic or theranostic PET tracer, the in vivo performance, (i.e., its actual binding, accumulation, and retention at the target site) should be thoroughly validated, particularly for the field of neuro-oncology, as CNS tumors show great inter- and intra-tumoral heterogeneous BBB and blood–tumor barrier permeability [17]. Furthermore, they are equipped with active efflux mechanisms [18] and a tumor microenvironment that has been shown to thwart the effectiveness of therapeutic compounds [19]. To come to objective performance parameters, quantification of the tissue uptake of tracers is key. In general, semi-quantitative parameters are used instead of performing true quantification based on metabolite analysis and thorough compartmental and kinetic modeling. For semi-quantitative analysis, the ratio of activity per unit volume of a region of interest (ROI) to the activity per unit whole body volume is calculated resulting in a standardized uptake value (SUV). This semi-quantitative parameter includes the total injected dose of radioactivity and compensates for patient size and passed time post-injection. The calculation of SUVs facilitates the comparison of uptake between tumor and healthy tissue, as well as between patients [20]. Non-specific tracers have (relatively) low tumor SUVs compared to healthy tissue SUVs (i.e., background) resulting in low tumor-to-background ratios (TBRs) and limited contrast. Tumor selective tracers show high TBRs, but in various degrees.

Today, PET already serves a broad applicability, which in the future will be expanded even further. The next paragraphs provide an overview of all PET traces that have thus far been applied for diagnostic and theranostic purposes in patients with CNS tumors. The applications of each tracer (many of which strongly correspond with the FDA’s definition of a companion or complementary diagnostic device), and the (dis)advantages will be discussed, with special attention for the added value of in vivo patient selection, drug selection, scheduling, and delivery over currently approved ex vivo diagnostic devices.

## 3. PET Tracers for Diagnostic Imaging of CNS Tumors

This paragraph describes all PET tracers that thus far have been used for diagnostic purposes in patients with CNS tumors (Table 1). Many of the diagnostic tracers are not very specific. However, some are sufficiently specific to potentially serve theranostic purposes, but are—for now—classified as diagnostic tracers because a therapeutic analog has not yet been developed. For the sake of clarity, the diagnostic CNS tracers will be arbitrarily subdivided based on their type of tumor target, being (i) abnormal metabolic processes present in cancer cells, (ii) increased perfusion shown in areas of disease, or (iii) the overexpression of (membrane) proteins.

### 3.1. Diagnostic Imaging of Abnormal Metabolic Processes

Among all diagnostic PET tracers, the most widely studied and used is [^18^F]-2-fluoro-2-deoxy-D-glucose ([^18^F]FDG). As an analog of glucose, [^18^F]FDG is taken up into cells by physiological glucose transport, after which it is phosphorylated to [^18^F]FDG-6-phosphate. Being slightly different from glucose, the phosphorylated product (FDG-6-P) is not metabolized further and remains trapped in the cell [21]. Increased cellular metabolism (i.e., glucose uptake and phosphorylation) is the underlying mechanism of higher SUVs of [^18^F]FDG in cancer cells compared to healthy tissue (i.e., Warburg effect) [21,22]. However, in particular in neuro-oncology studies, optimization of the TBR (i.e., contrast) is challenging because of the high physiological uptake of [^18^F]FDG in normal active brain tissue. To reduce uptake in healthy brain, complete fasting for a minimum of 6 h before the scan is recommended, and patients should be kept blindfolded in a quiet room during the uptake phase, or scanning times should be delayed [23]. With applying these procedures, [^18^F]FDG has shown acceptable sensitivity for the identification of anaplastic abnormalities in patients with low-grade glioma (LGG), high-grade glioma (HGG; Figure 1a [24]), CNS lymphoma, brain metastases, and meningioma [25,26,27]. Moreover, higher [^18^F]FDG uptake has shown a positive correlation with higher histologic grade and worse prognosis [26,28], making [^18^F]FDG PET a suitable tool for the prediction of progression-free survival (PFS) and overall survival (OS) [29]. [^18^F]FDG PET has shown not to be suited for radiation treatment planning, since the region of increased uptake generally covers a smaller volume than what was defined as malignant on T1-weighted gadolinium and T2-weighted MR-images [30]. In addition, [^18^F]FDG cannot distinguish between radiation-induced necrosis, changes in the tissue due to surgery, inflammation, or (remnant/recurrent) disease, making [^18^F]FDG not valuable for follow-up studies of CNS tumor therapies [21,22].

Next to [^18^F]FDG, amino acid PET tracers are increasingly applied, namely L-[methyl-^11^C]-methionine ([^11^C]Met), O-(2-[^18^F]-fluoroethyl)-L-tyrosine ([^18^F]FET) (Figure 1b), 3,4-dihydroxy-6-[^18^F]-fluoro-L-phenylalanine ([^18^F]DOPA), and 4-[^18^F]F-(2S,4R)-fluoroglutamine ([^18^F]FGln). Amino acid PET imaging is based on the overexpression of amino acid transporters and displays increased cell metabolism and cell division in most CNS tumor types (see Table 1) [21,22,31,32,33,34,35,36,37,38]. The use of amino acid tracers is advantageous over [^18^F]FDG because of the relatively low amino acid uptake in normal brain tissue, which results in higher contrast (i.e., TBR for [^18^F]DOPA 2.3 +/− 0.51, compared to 1.03 +/− 0.64 for [^18^F]FDG) [39]. Another advantage of amino acid tracers is the fact that their uptake does not depend on BBB permeability, as the barrier naturally possesses amino acid transporters. This enables the visualization of anaplastic regions even in areas that are not enhanced by contrast agents on MRI and where the BBB is assumed to be intact [7,40]. Amino acid PET imaging has shown to be promising for diagnostic purposes, as well as for biopsy and resection surgery planning [41,42,43]. Amino acid tracers also perform particularly well in differentiating early tumor progression from pseudoprogression (i.e., imaging changes that mimic a progressive tumor, but that are actually due to other causes, most commonly inflammation related to therapy), with significant higher SUVs or TBRs in (recurrent) LGG and HGG, and brain metastases compared to lesions with pseudoprogression [44,45,46]. The ability to recognize pseudoprogression, or treatment-related changes, is of high clinical value as it directs clinical decision making [47].

One of the first amino acid tracers used was [^11^C]Met [48]. [^11^C]Met uptake is mediated by the amino acid transporter LAT1 and has been widely used for the detection of CNS tumors, including glioma, germinoma, CNS lymphoma, CNS metastases, meningioma, mixed neural/glial tumors, and central neurocytoma [49,50,51,52,53]. Unlike [^18^F]FDG, [^11^C]Met is also useful in the follow-up of treatment response, as the decreased uptake of [^11^C]Met over time after therapy (i.e., surgery, chemo- or radiotherapy, or a combination of these) has shown to correlate with long-term survival [54]. To avoid the practical challenges related to the short half-life of carbon-11, next-generation amino acid tracers were labeled with fluorine-18, namely [^18^F]DOPA, [^18^F]FET, and [^18^F]FGln [22,29,55]. [^18^F]DOPA and [^18^F]FET are also taken up by the tumor via amino acid transporters located on the cell surface (i.e., LAT1 and LAT1 and LAT2, respectively) but unlike [^11^C]Met, they are not subsequently incorporated into proteins [40]. Both [^18^F]DOPA and [^18^F]FET show specific uptake in tumor cells, resulting in good contrast with similar predictive values as [^11^C]MET for diagnosing CNS tumors [44,56]. [^18^F]DOPA has also been shown to enable the monitoring of treatment response after anti-angiogenic therapy with bevacizumab [57]. [^18^F]FGln uptake is mediated by a different type of amino acid transporter (i.e., ATB(0)) and is subsequently metabolized to form glutamate, which is used for energy production [55]. In glioma patients, [^18^F]FGln has shown to be useful for differentiation between progressive and stable disease and enables the non-invasive delineation of tumors [37]. A study including three patients with brain metastases showed a high detection rate with the use of [^18^F]FGln compared to [^18^F]FDG (81.6% versus 36.8%, respectively) [38].

Another category of metabolic tracers is based on hypoxia, which is often present in cancer owing to the fact that rapidly growing tumor cells outgrow their blood supply. The presence of hypoxia has shown to be a poor prognostic factor for survival as it induces resistance to radiotherapy (i.e., hypoxia-induced radioresistance) [58]. PET offers a non-invasive tool to identify hypoxic areas or map oxygenation within CNS tumors, which potentially has great clinical impact by providing avenues for treatment adaptation before and during radiotherapy. 1-(2-Nitro-imidazolyl)-3-[^18^F]fluoro-2-propanol ([^18^F]FMISO) is the most widely studied hypoxia tracer [59]. Upon diffusion into cells, this derivative of nitroimidazole gets metabolized by nitroreductase enzymes at low oxygen levels [59]. By binding these nitroreductase enzymes, the tracer gets trapped in hypoxic viable cells (not in necrotic cells), which can be visualized by PET imaging (Figure 1d). A first study in three glioblastoma (GBM) patients reported a specific uptake of [^18^F]FMISO in tumor cells and not in healthy brain tissue [60]. [^18^F]FMISO also showed to be valuable for differentiation between LGG and GBM, with lesion to cerebellum uptake ratios of 1.22 ± 0.06 and 2.74 ± 0.60, respectively [61]. Subsequent studies showed a correlation between both uptake intensity and uptake volume and survival after radiotherapy, where higher intensity and larger volumes corresponded with worse survival [62,63,64]. A major limitation in the use of [^18^F]FMISO is its lipophilicity. Although this facilitates passage over an intact BBB, it also leads to a high background signal in the brain, as the clearance from plasma is slow. Nevertheless, compared with [^18^F]FDG, the sensitivity and specificity of [^18^F]FMISO are higher (i.e., 100 and 100%, compared to 100 and 66% for [^18^F]FDG, respectively) [61]. Still, other hypoxia tracers were developed that are less lipophilic, such as [^18^F]fluoroazomycin arabinoside ([^18^F]FAZA) [65,66] and 1-[2-[^18^F]Fluoro-1-(hydroxymethyl)-ethoxy]methyl-2-nitroimidazole ([^18^F]FRP-170) [67,68]. These hypoxia tracers have shown similar results compared to [^18^F]FMISO, but with better contrast in patients with GBM. To date, hypoxia tracers that contain therapeutic compounds have not yet been developed.

Finally, less routinely used diagnostic tracers that make use of abnormal metabolic processes include 3′-deoxy-3′-[^18^F]fluorothymidine ([^18^F]FLT), [^11^C]choline or [^18^F]fluorocholine, [^11^C]acetate, and ^62^Cu- or ^64^Cu-labeled tracers, which all show promising results for diagnosing patients with CNS tumors (Table 1) [21,22,29,69,70,71,72,73,74,75]. [^18^F]FLT displays the activity of thymidine kinase 1 (TK1), which is a cytosolic enzyme that is active during S and G2 cell cycle phases [21]. [^18^F]FLT herewith visualizes enhanced cell proliferation in cancer cells compared to normal brain tissue, resulting in good contrast and high sensitivity for the detection of HGG (Figure 1e) [69,76]. Most studies found [^18^F]FLT uptake only in areas of contrast enhancement on MRI, suggesting a dependency on BBB disruption [21,29]. One study by Jacobs et al., on the other hand, showed an uptake of [^18^F]FLT also in non-enhancing tumor areas and in LGG. The differences in uptake between HGG and LGG observed in this study suggest the potential of [^18^F]FLT to differentiate between HGG and LGG [76].

Choline tracers such as [^11^C]choline [70] and [^18^F]fluorocholine [77] have shown promising results for the identification and delineation of various CNS tumors, including LGG, HGG, and extra-axial tumors such as meningioma and schwannoma [78,79,80]. Choline is a precursor for the synthesis of phospholipids. In proliferating cancer cells, the endogenous synthesis of phospholipids is upregulated to build the cell membranes of the daughter cells. Therefore, proliferating cells show increased uptake of (radiolabeled) choline compared to surrounding healthy tissue (Figure 1c). Some studies reported false positive (5/110; 4.55%), false negative results (4/110; 3.64%), and an accuracy of 93/110 (84.5%) compared to 78/110 (70.9%) for [^18^F]FDG [70,79,81]. In addition, choline tracer uptake was shown to correspond only with areas of contrast enhancement on MRI, indicating dependency on BBB disruption, which limits the use of these tracers [70]. Similar to choline, acetate can be labeled with carbon-11 to depict increased phospholipid synthesis in glioma, CNS metastases, meningioma, and schwannoma [74,75,82,83]. The use of [^11^C]acetate for the detection of HGG showed a sensitivity of 90% compared to sensitivities of 100% and 40% for [^11^C]Met and [^18^F]FDG, respectively [73]. [^11^C]acetate also performed well in differentiation between HGG and LGG [72], and even between grade IV and grade III gliomas [75]. However, in meningioma, tumor grading with [^11^C]acetate was less valuable [74]. Lastly, facilitated by the overexpression of copper transport receptors, the visualization of increased copper uptake in cancer cells compared to normal brain tissue has shown promising results for CNS tumor diagnosis [71]. Copper is taken up into cells by the transporter receptor Ctr1 and distributed in different organelles and, in the case of tumor cells, also into the nucleus where copper is incorporated into Cu-dependent so-called cuproenzymes that are indispensable for upregulated electron and oxygen transportation [84]. An increased uptake of the copper tracer [^64^Cu][CuCl_2_] (i.e., copper chloride) has been shown in patients with GBM [85]. Another copper tracer [^62^Cu]-diacetyl-bis(*N^4^*-methylthiosemicarbazone) ([^62^Cu][Cu(ATSM)]) was originally developed as a hypoxia tracer for localizing malignant tumor tissue, but also appeared to be a promising predictor of both PFS and OS in patients with WHO grade II–IV glioma [86]. [^62^Cu][Cu(ATSM)] could theoretically also serve as a theranostic PET tracer for [^67^Cu][Cu(ATSM)] radionuclide therapy, as it showed significant selectivity for hypoxic tumor tissue, but the variable (low) uptake shown in the first in human studies demands further research into the usability of Cu-labeled tracers for targeted therapy in CNS tumors. The limited availability of copper isotopes is also hampering the development of copper-based theranostic tracers.

### 3.2. Diagnostic Imaging of Increased Perfusion

Apart from abnormal metabolic processes, CNS tumors also show increased perfusion, which can be visualized using [^13^N]ammonia ([^13^N]NH_3_). The use of [^13^N]NH_3_ to study cerebral blood flow has been performed in patients with glioma, brain metastases, and meningioma [87,88,89,90,91]. Xiangsong et al. showed increased tracer uptake to be present in 95% of MR contrast-enhancing lesions, suggesting BBB dependency [88]. No uptake was seen in non-neoplastic lesions rendering 100% specificity for CNS tumors in this study, including patients with LGG, HGG, and brain metastases from non-CNS tumors. Another study reported a sensitivity of 63% for the detection of LGG and HGG [89]. [^13^N]NH_3_ also showed to be useful for differentiation between recurrent cerebral astrocytoma and radiation necrosis (with moderate to high uptake compared to no uptake, respectively) [91]. In meningioma, similar results were found with good contrast to normal brain. However, differentiation between benign and atypical meningioma appears difficult [90].

PET-guided visualization of (increased) perfusion in CNS tumors can also be obtained using [^15^O]H_2_O, as shown by Bruehlmeier et al., who compared the kinetics of [^15^O]H_2_O to the kinetics of [^18^F]FMISO in patients with GBM in order to show that the level of [^18^F]FMISO uptake is independent of tumor perfusion [92,93].

### 3.3. Diagnostic Imaging of Upregulated (Membrane) Receptors

Finally, PET tracers have been directed to the translocator protein (TSPO), which is a target originating from inflammation research [94]. TSPO is a mitochondrial membrane protein that functions as a peripheral benzodiazepine receptor and is known for its expression in peripheral tissue, microglia, and astrocytes [95,96]. Its expression has shown to be upregulated in astrocytic tumors, and the targeting of TSPO with a radioactively labeled ligand, R-[^11^C]PK11195, showed potential for the detection of early anaplastic transformation in glioma [94,97,98,99,100]. One of the studies, including LGG and HGG of different subtypes (mainly astrocytoma and oligodendroglioma), showed that dynamic PET imaging using R-[^11^C]PK11195 can discriminate between low-grade astrocytoma and oligodendrogliomas [94]. Due to the short half-life of carbon-11 and the relatively low TBR provided by R-[^11^C]PK11195, a novel third generation ^18^F-labeled TSPO targeting ligand named [^18^F]GE-180 has been developed, which showed improved contrast in HGG patients [101,102].

## 4. PET Tracers for Theranostic Imaging of CNS Tumors

The above-mentioned tracers have shown added value for CNS tumor diagnostics and in some cases in follow-up studies. We here review PET tracers that serve the effective use of (radio)pharmaceuticals in patients with CNS tumors. Table 2 provides an overview of these theranostic PET traces.

### 4.1. PET Tracers for Guiding Targeted Radionuclide Therapy (i.e., Radiopharmaceuticals)

In 2010, a first study applying so-called peptide receptor radionuclide therapy (PRRT) for the treatment of CNS tumors was performed by Heute et al. [103]. The objective of PRRT is to target upregulated peptide receptors with a therapeutic radionuclide for localized therapy. As a first step, the expression and accessibility of a target peptide receptor, here the somatostatin receptor 2 (SSTR2), was studied by using an analog of somatostatin, octreotide, labeled with the positron-emitting radionuclide gallium-68 via a so-called DOTA chelator (i.e., [^68^Ga]Ga-DOTA-TATE, [^68^Ga]Ga-DOTA-TOC, or [^68^Ga]Ga-DOTA-NOC) [104]. When subsequent PET-imaging confirmed SSTR2 expression and accessibility by showing high uptake at the tumor site, patients were found eligible for targeted therapy using one of the DOTA peptides labeled with a therapeutic beta-particle-emitting radionuclide, either luthetium-177 or yttrium-90 [103,105,106,107]. The positron-emitting radionuclide can hereafter be used again for the assessment of therapeutic efficacy as shown in the first study using [^90^Y]Y-DOTA-TOC in patients with GBM, where post-therapeutic follow-up PET images showed a decreased uptake of [^68^Ga]Ga-DOTA-TOC, which is suggestive of therapeutic efficacy (Figure 2) [103]. Subsequent studies into PRRT for treatment of CNS tumors included patients with HGG and meningioma [105,106,107]. In meningioma, it was shown that the level of SSTR2 expression, measured by [^68^Ga]Ga-DOTA-TATE/-TOC uptake, correlates with the therapeutic efficacy of [^177^Lu]Lu-DOTA-TATE [105,106,107] or [^90^Y]Y-DOTA-TOC therapy [105]. In 2019, Verburg et al. performed a PET-guided drug delivery study aimed at optimizing the effect of PRRT in four patients with inoperable grade II meningioma [108]. Here, tumor uptake of [^68^Ga]Ga-DOTA-TATE upon intra-arterial (IA) versus intravenous (IV) administration was compared. Results showed a 2.7-fold higher tracer uptake after IA administration compared to IV administration. The potential benefit of IA PRRT for CNS tumors has further been underlined by a case report of a grade II meningioma patient treated with [^177^Lu]Lu-DOTA-TATE, showing a 79% decrease of tracer uptake on post-therapeutic [^68^Ga]Ga-DOTA-TOC images [109].

As of 2018, another target for PRRT, the transmembrane receptor neurokinin type-1, has been studied. By means of IHC, it was demonstrated that neurokinin type-1 receptor is highly expressed on glioma cells, where it facilitates mitogenesis, angiogenesis, cell migration, and the formation of metastases [110,111]. For theranostic purposes, neurokinin type-1 receptor can be targeted by [^213^Bi]Bi-DOTA-[Thi^8^, Met(O_2_)^11^]-substance P ([^213^Bi]Bi-DOTA-SP), an α-particle emitting radionuclide. Today’s literature comprises two studies investigating the therapeutic use of [^213^Bi]Bi-DOTA-SP in CNS tumors: one study including nine patients and a second study including 20 patients with recurrent GBM [112,113]. In both studies, patients received localized PRRT by an intracavitary injection of [^213^Bi]Bi-DOTA-SP with the co-injection of [^68^Ga]Ga-DOTA-SP for imaging purposes to assess pharmacokinetics and biodistribution. These analyses showed tracer uptake to be concentrated in the target lesions, with low systemic uptake (i.e., less than 5% of the total activity) in the kidneys and urine [112,113].

Another target for radionuclide therapy is fibronectin, which is not strictly a membrane receptor but rather a cell matrix protein involved in the adhesion and migration of cells [114]. In normal brain cells, the expression of fibronectin is downregulated upon maturation. However, in cancer cells, expression can be upregulated in order to promote angiogenesis [115]. In 2013, a PET-guided dose scheduling study was performed in six patients with inoperable brain and extracranial metastases from NSCLC or breast carcinoma. Eligible patients first received the human small immune protein L19SIP labeled with iodine-124, followed by PET imaging at 1, 4, 24, 48, and 96 h after administration to assess and predict the optimal protein dose for achieving potentially effective radiation at the target site(s) and to study unwanted accumulation of tracer at other sites (i.e., bone red marrow and healthy organs) to anticipate toxicity [114]. Upon PET-guided drug titration, patients received the therapeutic format [^131^I]I-L19SIP for treatment purposes.

Since 2017, prostate-specific membrane antigen (PSMA) has attracted wide interest, which is a target that is intensively and successfully exploited for the diagnosis and therapy of prostate cancer [116]. As shown by IHC of CNS tissues, PSMA is expressed in the tumor vasculature of glioma, breast cancer metastases, schwannomas, and peripheral nerve sheath tumors, while being absent in normal vessels [117,118,119]. The first PET studies investigating PSMA expression in CNS tumors made use of [^68^Ga]Ga-HBED-CC-PSMA ([^68^Ga]Ga-PSMA-11) and confirmed selective target expression and target accessibility in LGG, HGG, and gliosarcoma [120,121,122,123], with higher expression in HGG compared to LGG [120]. Interestingly, in undefined brain lesions, PSMA-PET imaging also showed diagnostic potential, as lesions with high [^68^Ga]Ga-PSMA-11 uptake were histopathologically confirmed to be glioma, atypical meningioma, and lymphoma [121]. Another study using 2-(3-(1carboxy-5-(6-[^18^F]fluoro-pyridine-3-carbonyl)-amino]-pentyl)-ureido)-pentanedioic acid ([^18^F]DCFPyL) in three GBM patients also confirmed specific target binding [124]. Finally, PSMA target expression has been studied using the anti-PSMA minibody, IAB2M, labeled with zirconium-89, [^89^Zr]Zr-IAB2M, in patients with HGG and metastatic brain tumors [125]. The selective target expression and target accessibility shown in these studies indicate the potential of PSMA as a target for the local delivery of therapeutic radionuclides [119]. The therapeutic efficacy of this approach for CNS tumors is hitherto only shown in patients with cerebral metastasis from prostate cancer that were treated with the anti-PSMA-based radiopharmaceuticals [^177^Lu]Lu-PSMA-617 and [^225^Ac]Ac-PSMA-617 [126,127,128]. These studies showed significant regression in the size of the cerebral tumors. Unfortunately, [^225^Ac]Ac-PSMA-617 also showed significant toxicity to the salivary glands due to physiologic PSMA expression [127,128,129,130]. Nevertheless, PSMA-PET is a promising theranostic tool for guiding the localized therapy of CNS tumors.

### 4.2. PET Tracers for Guiding Targeted Drug Therapy (i.e., Pharmaceuticals)

With increasing knowledge of cancer biology, crucial molecular targets have been identified, leading to the development of a wide range of targeted therapeutic pharmaceuticals, of which the receptor tyrosine kinase inhibitory drugs are the most rapidly expanding class [16,131]. Receptor tyrosine kinases (RTKs), including epidermal growth factor receptor (EGFR), vascular endothelial growth factor receptor (VEGFR), and platelet-derived growth factor receptor (PDGFR), among others, are transmembrane proteins that have been shown to serve a pivotal role in cell growth-related signal transduction. RTKs consist of an extracellular domain, which can be targeted with monoclonal antibody (mAb)-based inhibitory drugs, while the intracellular signal cascades can be inhibited by small molecule tyrosine kinase inhibitors (TKIs) via competition with adenosine triphosphate (ATP) [132].

#### 4.2.1. PET Imaging of Monoclonal Antibody-Based Inhibitory Drugs

In 2010, a first PET imaging study for CNS tumor targeting was performed using the mAb trastuzumab labeled with zirconium-89, [^89^Zr]Zr-trastuzumab, in patients with brain metastases from human epidermal growth factor receptor 2 (HER2)-positive breast cancer [133]. Results showed an 18-fold higher uptake in tumors than in normal brains, which is highly interesting given that it is generally believed that intact antibodies such as trastuzumab cannot pass the BBB. These findings suggest local BBB disruption and support the use of trastuzumab therapy in these patients. Another study using [^64^Cu]Cu-DOTA-trastuzumab in patients with brain metastases from breast cancer have confirmed the passage of trastuzumab over the BBB and the potential of PET to identify HER2-positive lesions non-invasively [134]. Lastly, for patients with brain metastases from breast cancer, the mAb pertuzumab was labeled with zirconium-89 forming [^89^Zr]Zr-pertuzumab [135]. Similar results were found as with trastuzumab, demonstrating safe and successful HER2 targeting.

In 2017, a first monoclonal antibody PET imaging study in children was performed in patients with diffuse midline glioma (DMG, formerly known as diffuse intrinsic pontine glioma (DIPG)). The target used for imaging in this study is vascular endothelial growth factor (VEGF) [136]. VEGF is a signaling protein that is located on the cell surface of endothelial cells, which promotes angiogenesis. VEGF can be targeted using the monoclonal antibody bevacizumab (Avastin). Tumor uptake and biodistribution of the drug was visualized by means of PET imaging with [^89^Zr]Zr-bevacizumab [137]. The results showed that ^89^Zr-bevacizumab PET imaging is feasible and safe in pediatric patients above 6 years of age. Among seven patients, a marked inter- and intratumoral heterogeneity of ^89^Zr-bevacizumab uptake was observed, suggesting considerable variability in the tumor delivery of bevacizumab in DIPG patients, which was possibly related to the heterogeneous expression of VEGF and/or variable integrity of the BBB (Figure 3) [137].

#### 4.2.2. PET Imaging of Tyrosine Kinase Inhibitors

TKIs can be labeled with carbon-11 and sometimes fluorine-18 to enable an in vivo assessment of pharmacokinetics and biodistribution. In the first proof-of-principle clinical trials, PET with radiolabeled TKIs (TKI-PET) showed great value for the prediction of therapy response and possible toxicity in patients with brain metastases from NSCLC or breast cancer [16]. In 2011, a case report was published describing the use of [^11^C]C-erlotinib (Tarceva) to study the target expression of EGFR and binding potential of erlotinib in a patient with brain metastases from NSCLC. PET/CT showed accumulation of the tracer in multiple brain metastases, and post-treatment MRI demonstrated regression of the enhancing lesions upon treatment with erlotinib [138]. Another TKI-PET tracer that has been used in CNS tumors is [^11^C]C-lapatinib, which was used to study the target expression and accessibility of HER2 in three patients with brain metastases from (HER2-positive) breast cancer [139]. An uptake of [^11^C]C-lapatinib was observed in tumor tissue and not in normal brain, and the PET procedures were well tolerated by the patients. Lastly, Varrone et al. assessed the TKI-PET tracer [^11^C]C-osimertinib to evaluate the brain accessibility and biodistribution of osimertinib [140]. Osimertinib is a drug that is known for its efficacy in patients with brain metastases, and it was here studied with PET in eight healthy volunteers with an intact BBB. The results demonstrated the rapid and high uptake of osimertinib in the brain, highlighting the potential of this TKI and warranting further research in CNS tumor patients.

## 5. Final Considerations and Future Direction

This review summarized a significant list of PET tracers currently available for diagnostic and, even more appealing, for theranostic purposes in CNS tumors. Out of these, radioactively labeled amino acids and (other) small molecules are of particular interest due to their non-dependency on BBB disruption to reach CNS tumor cells, together with their excellent tumor-to-background contrast. Other tracers such as monoclonal antibodies are more specific but also more dependent on BBB disruption, which at this moment diminishes their therapeutic potential.

Attempts to overcome an intact BBB for improved drug delivery have a long history in which chemical disruption via the hyperosmotic solution mannitol was the first, followed by applying other drugs that influence passive diffusion or active transport mechanisms [141,142]. In another approach, therapeutic pharmaceuticals are attached to molecules naturally transported across the barrier (i.e., viral vectors, nanoparticles, liposomes, exosomes), and by using transporter or receptor (e.g., transferrin receptor) ligands, such as how transport is done with an anti-amyloid-beta antibody modified into a bispecific format with the capacity to undergo transferrin receptor 1 (TfR1)-mediated transcytosis for Alzheimer’s disease [131]. Other attempts for improved drug delivery include mechanical disruption techniques such as by applying radiotherapy, microwaves, or ultrasound (e.g., high-intensity focused ultrasound, HIFU), or by using novel technical approaches such as stereotaxic injection into the cerebrospinal fluid or convection-enhanced delivery (CED). However, hardly any of these historic attempts to improve the efficacy of CNS tumor therapies have made use of molecular PET imaging to visualize or quantify the effect on the BBB and the actual passage of pharmaceuticals, although since the 1980s, several PET imaging studies into manipulation of the BBB passage have been performed [141,143,144,145,146]. Recently, Lesniak et al. were the first to elegantly demonstrate the added value of PET imaging for studying the BBB passage of biological pharmaceuticals in mice by using [^89^Zr]Zr-bevacizumab to compare brain uptake upon IA and IV administration in combination with BBB opening via mannitol [147,148]. By combining IA administration with BBB opening, >10 times higher monoclonal antibody delivery to the brain could be obtained. A follow-up study using ^89^Zr-labeled polyamidoamine dendrimers showed only a marginal improvement of brain delivery upon IA administration combined with BBB opening, while the similar approach for ^89^Zr-labeled nanobodies showed a 2.5-fold increase of brain uptake [149]. The results of Lesniak et al., indicating the advantage of IA administration, are in line with the results of the previously discussed PET imaging studies using [^177^Lu]Lu-DOTA-TATE in patients with grade II meningioma [108,109]. We emphasize that this so-called theranostic application of PET could be of great value for the visualization of the effect of attempts to overcome the BBB and increase the efficacy of targeted therapies, which is not only relevant for effective treatment of brain tumors but also for the detection and treatment of other brain diseases.

## 6. Conclusions

Although morphological MRI is the current standard of care for the diagnostic workup, treatment planning, response monitoring, and surveillance during follow-up of CNS tumors, we envision that PET imaging will gain more and more ground in the next years because of the rapid development of promising diagnostic and especially theranostic tracers [47,150,151]. Molecular PET imaging enables non-invasive CNS tumor diagnostics, which is potentially useful for studies into cancer genetic diversity and cancer evolution over time and space. Molecular PET imaging also has the potential to serve as companion or complementary diagnostic (i.e., theranostic) tool to improve the therapeutic efficacy of targeted pharmaceuticals and radiopharmaceuticals as well as ensure greater patient safety, e.g., by selecting more effective drugs and appropriate patient groups and by optimizing drug delivery strategies. While these developments hold great promise for the personalized care of patients, they are also attractive for pharma companies as they will increase the success rate in drug development, shorten the time to market, reduce the number of patients needed in clinical trials, and therefore reduce costs. Particularly with the emergence of simultaneous dual-modality scanners (notably PET-MRI), the diagnostic and therapeutic applicability in CNS tumors will be largely expanded, as this technique provides both exquisite structural characterization by MRI and highly specific functional characterization and objectifiable targeted treatment options by PET. Especially in patients with CNS tumors, this non-invasive approach likely facilitates and accelerates future drug development and drug delivery studies, possibly leading to improved quality of life for patients and more cost-effective care.

## Figures and Tables

**Figure 1 ijms-21-01029-f001:**
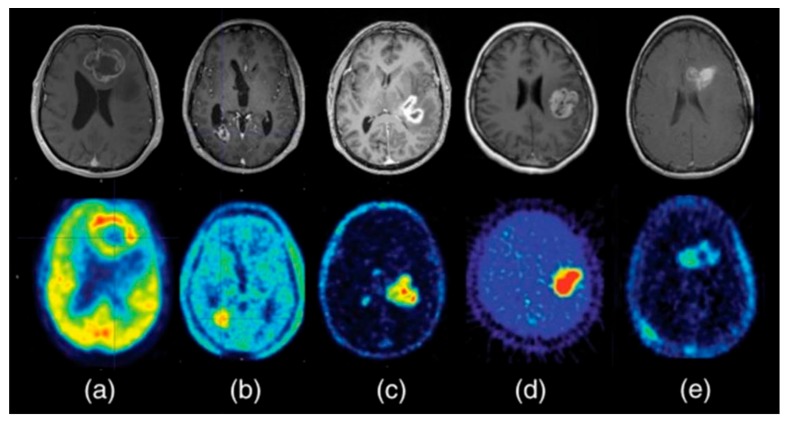
Contrast-enhanced Magnetic Resonance Imaging (MRI) (top row) and multiple PET tracers for diagnostic imaging (bottom row) in glioblastoma. (**a**) [^18^F]-2-fluoro-2-deoxy-D-glucose ([^18^F]FDG), (**b**) O-(2-[^18^F]-fluoroethyl)-L-tyrosine ([^18^F]FET), (**c**) [^18^F]Fluorocholine, (**d**) 1-(2-Nitro-imidazolyl)-3-[^18^F]fluoro-2-propanol ([^18^F]FMISO), (**e**) 3′-deoxy-3′-[^18^F]fluorothymidine ([^18^F]FLT). Adapted from [24]. This research was originally published in *Glioblastoma [internet]*. Bolcaen, J.; Acou, M.; Descamps, B.; Kersemans, K.; Deblaere, K.; Vanhove, C.; Goethals, I. PET for therapy response assessment in glioblastoma. In *Glioblastoma [Internet]*; De Vleeschouwer, S., Ed.; Codon Publications: Brisbane, AU, 2017.

**Figure 2 ijms-21-01029-f002:**
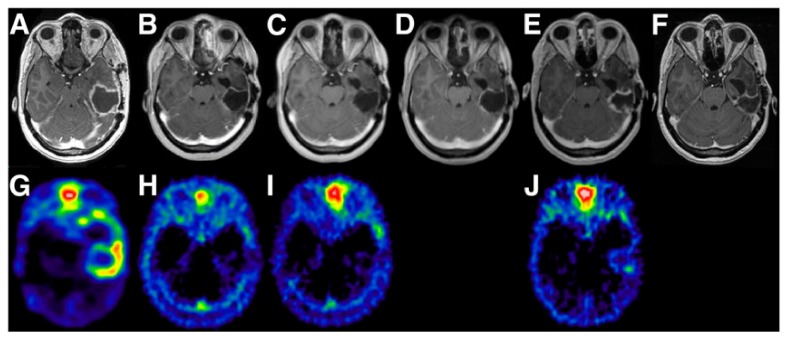
Response of high-grade glioma to local therapy with a cumulated 2.2-GBq dose of [^90^Y]Y-DOTA-TOC given in three cycles (from left to right: study before therapy, control study 3 months after second dose, control study 3 months after third dose, and control study 23 months after third dose). (**A**–**F**) T1-weighted enhanced MR images show diminishing contrast agent in tissue surrounding resection cavity throughout therapy. (**A**–**J**) [^68^Ga]Ga-DOTA-TOC PET images representing somatostatin receptor status show increased tracer uptake around resection cavity before therapy (**G**) and normalization in control studies (**A**–**J**). Adapted from [103]. This research was originally published in JNM. Heute D, Kostron H, von Guggenberg E, Ingorokva S, Gabriel M, Dobrozemsky G, et al. Response of recurrent high-grade glioma to treatment with (90)Y-DOTATOC. J Nucl Med. 2010;51(3):397-400. © SNMMI.

**Figure 3 ijms-21-01029-f003:**
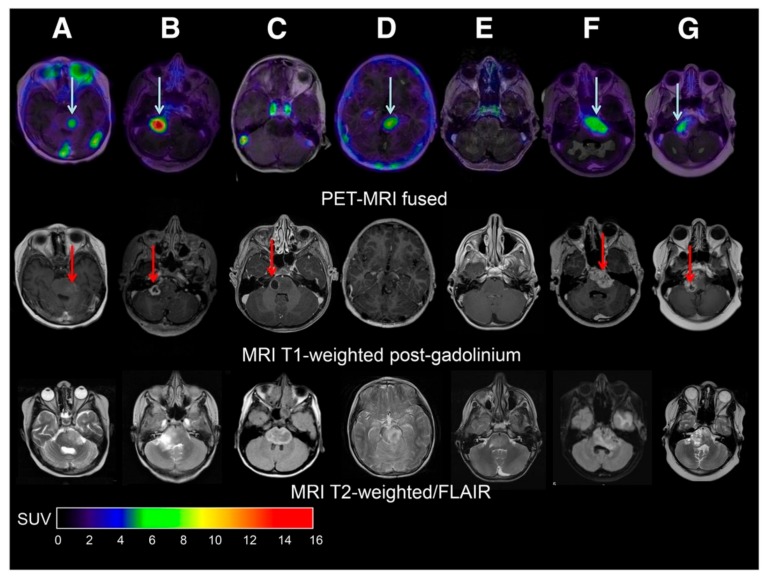
MRI and PET-MRI fusion images of patients with diffuse midline glioma (DMG). (**A–G**) Top row: [^89^Zr]Zr-bevacizumab PET (144 hrs post-injection) fused with T1-Gadolinium (Gd) weighted MRI per patient; middle row: T1-Gd weighted MRI; lower row: T2-weighted/Fluid-attenuated inversion recovery (FLAIR) MR-images. Five tumors show a variable uptake of [^89^Zr]Zr-bevacizumab (white arrows), with both PET negative and positive areas within each tumor. Two primary tumors are completely PET negative (**C** and **E**), while the T2 weighted images show tumor infiltration in the whole pons of both patients. In the middle row, the red arrows represent the areas of contrast enhancement within the tumor. In four out of five primary tumors, the PET-positive area corresponds with the contrast-enhancing area on MRI of the tumors (**A**,**B**,**F** and **G**). In **C**, the tumor shows an MRI contrast-enhancing area, while there is no ^89^Zr-bevacizumab uptake. Figure **D** shows a PET-positive tumor, while no Gd-enhancement is observed on MRI. Adapted from [137]. This research was originally published in JNM. Jansen MH, Veldhuijzen van Zanten SEM, van Vuurden DG, Huisman MC, Vugts DJ, Hoekstra OS, et al. Molecular Drug Imaging: (89)Zr-Bevacizumab PET in Children with Diffuse Intrinsic Pontine Glioma. J Nucl Med. 2017;58(5):711-6. © SNMMI.

**Table 1 ijms-21-01029-t001:** Applications of Positron Emission Tomography (PET) tracers for diagnostic imaging of Central Nervous System (CNS) tumors.

Tracer	Target	CNS Tumor Types	References
[^18^F]FDG	Elevated glucose metabolism	glioma, CNS lymphoma, CNS metastases, meningioma, DMG	[21,22,23,24,25,26,27,28,29,30]
[^11^C]Met	Increased amino acid uptake	glioma, germinoma, CNS lymphoma, CNS metastases, meningioma, mixed neural/glial tumors, central neurocytoma	[21,22,31,41,44,46,47,49,50,51,52,53,54,56]
[^18^F]FET	Increased amino acid uptake	glioma, CNS lymphoma, CNS metastases, meningioma, medulloblastoma, DMG	[21,22,39,41,43,44,45,46,47]
[^18^F]DOPA	Increased amino acid uptake	glioma, CNS metastases, meningioma	[21,22,34,40,41,46,47,56,57]
[^18^F]FGln	Increased amino acid uptake	glioma, CNS metastases	[22,29,37,38,55]
[^18^F]FMISO	Cell metabolism under hypoxia	glioma	[59,60,61,62,63,64]
[^18^F]FAZA	Cell metabolism under hypoxia	HGG	[65,66]
[^18^F]FRP-170	Cell metabolism under hypoxia	HGG	[67,68]
[^18^F]FLT	Increased activity of thymidine kinase 1	HGG, CNS metastases, meningioma	[21,29,69,76]
[^11^C]choline	Increased synthesis of phospholipids	glioma, CNS metastases, meningioma, schwannoma	[70,79,80,81]
[^18^F]fluorocholine	Increased synthesis of phospholipids	glioma, CNS metastases, meningioma, schwannoma	[70,77,78]
[^11^C]Acetate	Increased amino acid uptake	glioma, CNS metastases, meningioma, schwannoma	[72,73,74,75,82,83]
[^64^Cu][CuCl_2_]	Increased copper uptake	HGG	[71,85]
[^62^Cu][Cu(ATSM)]	Increased copper uptake	HGG, CNS metastases, meningioma	[84,86]
[^13^N]NH_3_	Increased perfusion	glioma, CNS metastases, meningioma	[87,88,89,90,91]
R-[^11^C]PK11195	Upregulated TSPO	glioma	[94,97,98,99,100]
[^18^F]GE-180	Upregulated TSPO	HGG	[101,102]

Abbreviations: [^11^C]Met: L-[methyl-^11^C]-methionine, CNS: central nervous system, [^64^Cu][CuCl_2_]: [^64^Cu]chloride; [^62^Cu][Cu(ATSM)]: [^62^Cu]-diacetyl-bis(N4-methylthiosemicarbazone, DMG: diffuse midline glioma, [^18^F]FDG: [^18^F]-2-fluoro-2-deoxy-D-glucose, [^18^F]DOPA: 3,4-dihydroxy-6-[^18^F]-fluoro-L-phenylalanine, [^18^F]FAZA: [^18^F]fluoroazomycin arabinoside, [^18^F]FET: O-(2-[^18^F]-fluoroethyl)-L-tyrosine, [^18^F]FGln: 4-[^18^F]F-(2S,4R)-fluoroglutamine [^18^F]FLT: 3′-deoxy-3′-[^18^F]fluorothymidine, [^18^F]FMISO: 1-(2-Nitro-imidazolyl)-3-[^18^F]fluoro-2-propanol, [^18^F]FRP-170: 1-[2-[^18^F]Fluoro-1-(hydroxymethyl)-ethoxy]methyl-2-nitroimidazole, HGG: high-grade glioma, [^13^N]NH3: [^13^N]ammonia, TSPO: translocator protein.

**Table 2 ijms-21-01029-t002:** Applications of PET tracers for theranostic imaging of CNS tumors

Tracers	Target	CNS Tumor Types	Drug	References
**Tracers for radionuclide therapy**			**Radiopharmaceutical drug**	
[^68^Ga]Ga-DOTA-TATE	SSTR2	Meningioma	[177Lu]Lu-DOTA-TATE	[105,107,108,109]
[^68^Ga]Ga-DOTA-TOC	SSTR2	HGG, recurrent GBM, meningioma	[90Y]Y-DOTA-TOC	[103,105,106]
[^68^Ga]Ga-DOTA-SP	NK1-R	GBM and recurrent GBM	[213Bi]Bi-DOTA-SP	[112,113]
[^131^I]I-L19SIP	Fibronectin	CNS metastases	[131I]I-L19SIP	[114]
[^68^Ga]Ga-PSMA-11	PSMA	Glioma, CNS metastases, meningioma	[177Lu]Lu-PSMA-617^1^	[126]
			[225Ac]Ac-PSMA-617^1^	[127,128]
[^18^F]DCFPyL	PSMA	GBM	na	[124]
[^89^Zr]Zr-IAB2M	PSMA	HGG and CNS metastases	na	[125]
**Tracers for chemotherapeutic therapy**			**Pharmaceutical drug**	
*Tyrosine kinase inhibitors*				
[^11^C]C-erlotinib	EGFR	CNS metastases from NSCLC	Erlotinib	[138]
[^11^C]C-lapatinib	EGFR and HER-2	CNS metastases	Lapatinib	[139]
*Monoclonal antibodies*				
[^89^Zr]Zr-trastuzumab	HER2	CNS metastases	Trastuzumab	[133]
[^64^Cu]Cu-DOTA-trastuzumab	HER2	CNS metastases	Trastuzumab	[134]
[^89^Zr]Zr-bevacizumab	VEGF	Pediatric DMG	Bevacizumab	[137]
[^89^Zr]Zr-pertuzumab	HER2	CNS metastases	Pertuzumab	[135]

^1^ The radiopharmaceutical drug of PSMA-11 was only evaluated in cerebral metastases from castration-resistant prostate cancer [126,127,128]. Abbreviations: DCFPyL: 1carboxy-5-(6-[^18^F] fluoro-pyridine-3-carbonyl)-amino]-pentyl)-ureido)-pentanedioic acid, DMG: diffuse midline glioma, DOTA-SP: DOTA-[Thi8, Met(O2)11]-substance P, DOTA-TATE: DOTA-Tyr3-octreotate, DOTA-TOC: DOTA-Tyr3-octreotide, ECM: Extracellular matrix, EGFR: Epidermal growth factor receptor, GBM: glioblastoma, GPCR: G-protein coupled receptor, HER-2: human epidermal growth factor receptor 2, HGG: high-grade glioma, LGG: low-grade glioma, NK1-R: transmembrane neurokinin type-1 receptor, na: not available, PSMA: Prostate-specific membrane antigen, PSMA-11: HBED-CC-PSMA, SSTR2: somatostatin receptor type 2, VEGF: vascular endothelial growth factor, VEGFR: vascular endothelial growth factor receptor.

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
