# Peer review of "The Added Value of Diagnostic and Theranostic PET Imaging for the Treatment of CNS Tumors"

_ijms, 2020, doi:10.3390/ijms21031029_

Round 1

Reviewer 1 Report

Overview and general recommendation:

Today, diagnostic and theranostic approaches for CNS tumors have some limitations due to intrinsic characteristics of this tumor such as the high heterogeneity, the presence of BBB, the difficulty of access to the tumor mass. The development of new radiotracers for Positron Emission Tomography (PET) imaging could be very interesting for the diagnosis and the therapy of this type of tumor.

Here, the authors discuss the state-of-art of the PET tracers for diagnostic and theranostic imaging of CNS tumors.

I found the paper to be overall well written and discuss the state-of-art of the PET tracers.

I ask that the authors specifically address each of my comments in their response.

Major comments:

Page 6 line 239; hypoxia is often present in CNS tumors but the authors do not underline that it is important to have a non-invasive technique to identify the hypoxic areas within the tumor because they are associated to radio-resistance. The authors should edit this part. Page 7 line 264; authors discuss the major limitation of FLT but don’t discuss the ability of FLT to differentiate high-grade glioma and GBM from low-grade glioma compared to FET or MET. Authors should discuss it and add this reference: Jacobs AH et al J Nucl Med. 2005 Dec;46(12):1948-58. In paragraph 3.1 authors should add two recent radiotracers which are very interesting for glioma: 11C-acetate e 18F-glutamine. I suggest two references: Kim S et al. Eur J Nucl Med Mol Imaging. 2018 Jun;45(6):1012-1020; Venneti S et al. Sci Transl Med. 2015 Feb 11;7(274):274ra17.

Minor comments

I suggest changing the title of paragraph 2, page 2, line 92 Page 4 line 152, there are too many brackets after SUV.

Author Response

Today, diagnostic and theranostic approaches for CNS tumors have some limitations due to intrinsic characteristics of this tumor such as the high heterogeneity, the presence of BBB, the difficulty of access to the tumor mass. The development of new radiotracers for Positron Emission Tomography (PET) imaging could be very interesting for the diagnosis and the therapy of this type of tumor.

Here, the authors discuss the state-of-the-art of the PET tracers for diagnostic and theranostic imaging of CNS tumors.

I found the paper to be overall well written and discuss the state-of-the-art of the PET tracers.

I ask that the authors specifically address each of my comments in their response.

Thank you for your compliment and underlining of the relevance of the manuscript to the field of neuro-oncology. Below you will find our point-by-point response to your insightful comments.

Major comments:

Page 6 line 239; hypoxia is often present in CNS tumors but the authors do not underline that it is important to have a non-invasive technique to identify the hypoxic areas within the tumor because they are associated to radio-resistance. The authors should edit this part.

We agree that this information should be added when discussing hypoxia as a possible target for non-invasive diagnosis and treatment of CNS tumors. We have edited the relevant paragraph of our manuscript (page 6, line 254) and added the following text:

The presence of hypoxia has shown to be a poor prognostic factor for survival as it induces resistance to radiotherapy (i.e., hypoxia-induced radioresistance) [58]. PET offers a non-invasive tool to identify hypoxic areas or map oxygenation within CNS tumors, which potentially has great clinical impact by providing avenues for treatment adaptation before and during radiotherapy.

Page 7 line 264; authors discuss the major limitation of FLT but don’t discuss the ability of FLT to differentiate high-grade glioma and GBM from low-grade glioma compared to FET or MET. Authors should discuss it and add this reference: Jacobs AH et al J Nucl Med. 2005 Dec;46(12):1948-58.

We agree that the added value of [18F]FLT for differentiation of glial tumors should be added to the manuscript. We appreciate the suggested reference. We have supplemented the text with the requested information and compared it to [11C]Met. The study by Jacobs et al.did not allow for direct comparison with [18F]FET which was therefore not included. The following text is added (page 7, line 283):

[18F]FLT herewith visualizes enhanced cell proliferation in cancer cells compared to normal brain tissue, resulting in good contrast and high sensitivity for the detection of HGG (Figure 1e) [69,76]. Most studies found [18F]FLT uptake only in areas of contrast enhancement on MRI, suggesting dependency on BBB disruption [21,29]. One study by Jacobs et al., on the other hand, showed uptake of [18F]FLT also in non-enhancing tumor areas and in LGG. Differences in uptake between HGG and LGG observed in this study suggest potential of [18F]FLT to differentiate between HGG and LGG [76].

In paragraph 3.1 authors should add two recent radiotracers, which are very interesting for glioma: 11C-acetate e 18F-glutamine. I suggest two references: Kim S et al. Eur J Nucl Med Mol Imaging. 2018 Jun;45(6):1012-1020; Venneti S et al. Sci Transl Med. 2015 Feb 11;7(274):274ra17.

We agree that this information was lacking from chapter 3 of the manuscript. We appreciate the suggested references and have rewritten the chapter and edited the corresponding Table 1 based on these suggestions. The following texts have been added to the paragraphs on amino acid PET tracers (page 6 line 248) and the paragraph on less routinely used tracers (page 7, line 299), respectively:

[18F]FGln uptake is mediated by a different type of amino acid transporter (i.e., ATB(0)) and is subsequently metabolized to form glutamate used for energy production [55]. In glioma patients, [18F]FGln has shown to be useful for differentiation between progressive and stable disease and enables non-invasive delineation of tumor [37]. A study including three patients with brain metastases showed a high detection rate with use of [18F]FGln compared to [18F]FDG (81.6% versus 36.8%, respectively) [38].

Similar to choline, acetate can be labelled with carbon-11 to depict increased phospholipid synthesis in glioma, CNS metastases, meningioma and schwannoma [74,75,82,83]. Use of [11C]acetate for detection of HGG showed a sensitivity of 90%, compared to sensitivities of 100% and 40% for [11C]Met and [18F]FDG, respectively [73]. [11C]acetate also performed well in differentiation between HGG and LGG [72], and even between grade IV and grade III gliomas [75]. In meningioma, however, tumor grading with [11C]acetate was less valuable [74].

Minor comments:

I suggest changing the title of paragraph 2, page 2, line 92.

            Thank you for this suggestion; we changed the title of paragraph 2 to:

Advanced technology and applicability of molecular PET imaging for CNS tumors

Page 4 line 152, there are too many brackets after SUV.

            That is correct; we have deleted the unnecessary bracket after SUV.

Reviewer 2 Report

The review "The added value of diagnostic and theranostic PET imaging for the treatment of CNS tumors" by Pruis and co-authors is well written and interesting to read. It contains all necessary explanations to the topic and will be helpful for the research field and general auditory.

Author Response

The review “The added value of diagnostic and theranostic PET imaging for the treatment of CNS tumors” by Pruis and co-authors is well written and interesting to read. It contains all necessary explanations to the topic and will be helpful for the research field and general auditory.

Thank you for your compliments,and for underlining the added value of this manuscript to the research field.

Round 2

Reviewer 1 Report

The manuscript is well written and it has been improved.